# Association of Sociodemographic Factors with Tuberculosis Outcomes in Mississippi

**DOI:** 10.3390/diseases11010025

**Published:** 2023-02-01

**Authors:** Omer Osman, Azad R. Bhuiyan, Amal K. Mitra, Vincent L. Mendy, Sophia Leggett, Clifton Addison

**Affiliations:** 1Department of Natural Sciences and Environmental Health, Mississippi Valley State University, Itta Bena, MS 38941, USA; 2Department of Epidemiology and Biostatistics, School of Public Health, College of Health Sciences, Jackson State University, Jackson, MS 39213, USA; 3Department of Behavioral and Environmental Health, School of Public Health, College of Health Sciences, Jackson State University, Jackson, MS 39213, USA

**Keywords:** tuberculosis, Mississippi, sociodemographic factors

## Abstract

Tuberculosis (TB) is one of the leading causes of death worldwide. In the US, the national incidence of reported TB cases was 2.16 per 100,000 persons in 2020 and 2.37 per 100,000 persons in 2021. Furthermore, TB disproportionately affects minorities. Specifically, in 2018, 87% of reported TB cases occurred in racial and ethnic minorities in Mississippi. Data from TB patients from the Mississippi Department of Health (2011–2020) were used to examine the association between sociodemographic subgroups (race, age, place of birth, gender, homelessness, and alcohol use) with TB outcome variables. Of the 679 patients with active TB cases in Mississippi, 59.53% were Black, and 40.47% were White. The mean age was 46 ± ten years; 65.1% were male, and 34.9% were female. Among patients with previous TB infections, 70.8% were Black, and 29.2% were White. The rate of previous TB cases was significantly higher among US-born (87.5%) persons compared with non-US-born persons (12.5%). The study suggested that sociodemographic factors play a significant role in TB outcome variables. This research will help public health professionals to develop an effective TB intervention program that addresses sociodemographic factors in Mississippi.

## 1. Introduction

Tuberculosis (TB) is a communicable and reportable disease caused by the bacterium Mycobacterium tuberculosis [1]. According to a World Health Organization report, TB is the 13th leading cause of death and the second leading infectious killer disease after COVID-19 worldwide [2]. Of worldwide TB deaths, in 2018, the mortality rates comprised 90% of adults, of which 65% were male. Out of 10.4 million people with TB, 74% of cases were reported in Africa [3]. While the number of TB cases has been quite low in the United States (US), the Centers for Disease Control and Prevention (CDC) still reported over 9000 TB cases in the US in 2018 [4]. The national incidence rate of TB was 2.9 cases per 100,000 persons. 

TB disproportionately affects specific populations. According to [5] in 2018, 87% of all reported TB cases occurred in racial and ethnic minorities. African Americans are among one of the most highly susceptible groups and have a disproportionally higher incidence of the disease. While the rates of TB in African Americans have been cut in half (from 8.8 cases per 100,000 persons in 2008 to 4.4 in 2018) over the past decade, the rate of TB disease in African Americans is over eight times higher than the rate of TB in the White population (0.5 cases per 100,000). Importantly, TB is more prevalent in 25–44-year-old African Americans compared to White males of the same age group [6]. Several factors contribute to this high incidence, including but not limited to the increased incidence of poverty and homelessness among the target population. Moreover, the risk of contracting TB and its poor outcome among native-born African Americans exceeds the risks of the same outcomes in other groups. These factors translate to approximately three-to-seven-times higher morbidity rates among those who are not from a White background and account for almost 50% of all new cases of TB diagnosed in White people [6]. 

Several internal and external factors can influence the prevalence and mortality of TB, such as treatment outcomes, the onset of the disease, the overall health of the population, diabetes, chronic lung diseases, alcohol and tobacco use, and socioeconomic and environmental factors. Additionally, certain comorbid medical conditions contribute to disparities in the rate of TB. Recent studies have aimed to understand the persistent disparities that SES and an unequal burden of TB risk factors may be confounding. Our understanding of racial and ethnic disparities in TB in the US investigations of individuals’ social and geographic contexts hold clues for understanding the drivers of TB disparities in different contexts, as well as for designing context-specific interventions to ameliorate them [3]. Despite the long-standing observation that racial and ethnic minorities in the US have higher TB case rates than White people, the proportion of this elevated risk related to socioeconomic status (SES) has not been identified [6,7]. 

In alignment with the national incidents and factors of TB, there are discrepancies at the state level. Specifically, in 2018, the state of Mississippi had a case rate of 2.2 per 100,000 in 2019, which was lower than the national rate. However, Hinds County, Mississippi, had one of the highest rates of TB at 5.6 cases per 100,000 in 2019, which was much higher than the national rate. Additionally, the analysis of socio-demographics conducted at the county level by the Mississippi Department of Health (MSDH) identified an increase in TB cases among the homeless population from 12% in 2008 to 19.8% in the year 2009, along with the presence of the MS0006 strain in a large number of cases [8].

Mississippi saw a drop in the number of cases from 81 in 2018 to 57 in 2019 [9]. A newer strain of TB, initially known as PCR0025 and later renamed MS0006, happened to be the main agent of disease transmission in the state. Additionally, the analysis of socio-demographics conducted at the county level by the state department of health in Mississippi identified an increase in TB cases among the homeless population from 12% in 2008 to 19.8% in the year 2009, along with the presence of the MS0006 strain in a large number of cases [8].

The socioeconomic determinants of TB have long been recorded in historical accounts as well as in the scientific literature. Despite coordinated TB control efforts, inequalities in the incidence of TB associated with nativity (where a person is born), race, and SES persist [10]. Applying the social determinants of the health paradigm to infectious etiologies, especially TB, might shed light on more distant social and environmental variables that may be impeding our capacity to minimize TB inequalities. Many studies have found racial differences in TB incidence in the United States. However, few researchers have placed these differences within the broader context of socioeconomic and environmental determinants of health.

Therefore, the purpose of this study was to assess the association between sociodemographic characteristics, i.e., gender, age, race, country of birth, and homelessness status, and TB outcome variables, i.e., multidrug-resistant TB, site of TB, and previous TB infections in Mississippi.

## 2. Materials and Methods

The research questions for the study were as follows: RQ1: Is there an association between demographic subgroups (race, age, country of birth, and gender) and previous TB in Mississippi? RQ2: Is race associated with TB outcome variables (MDR-TB and TB site) adjusting for other covariates? RQ3: Are there differences between homeless and long-term care facility residents among TB outcomes in Mississippi?

A descriptive study of secondary data was conducted in which a significant sample of individuals was recruited, and their exposure and health outcomes were measured. Secondary data from 2011 to 2020 were collected from MSDH’s TB Control, which is dedicated to TB-related statistics and control. It records the health-/disease-related data, including, but not limited to, birth, death, cancer, and TB, which are available to the staff of the health department, researchers, and public health departments. During the analysis and preparation of the results, the purpose of the study and research objectives remained the main priority.

The participants of the study fulfilled the following inclusion criteria: 18 years of age or older with a confirmed TB case. The exclusion criteria were patients with other Mycobacterial infections who were aged less than 18 years.

Limitations: Since these data were not primarily intended for this research, an inherent risk was present that many values might be missing, which could lead to possibilities of bias and a lack of data on some confounders. However, the data analysis showed that only the age variable had seven missing values, which did not impact the present study.

Ethical approval: The study was approved by the Institutional Review Board of JACKSON STATE UNIVERSITY (protocol code 0023-22 on 17 June 2022).

### Operational Definitions of TB Outcome Variables

Previous TB: Recurrent or previous TB is defined by the CDC as TB that recurs due to reinfection or relapse after a patient is thought to be cured [4]. It is a challenge to TB control. The risk factors for recurrent TB after non-MDR-TB therapy were classified as host and bacillary risk factors. 

Secondary TB: Secondary TB is the gradual progression of primary TB or the reactivation of the old lesions of TB into a chronic form, causing secondary TB. Reinfection with TB is also termed secondary TB. Secondary TB is characterized by extensive tissue damage as a result of immunologic reactions between the bacilli and its host [11]. Between 90 and 95% of people with secondary TB obtain the disease by reactivation of the latent primary form, while the remaining percentage is caused by reinfection with Mycobacterium TB.

Latent TB Infection (LTBI): A total of 85% of TB disease cases in the US may be attributed to the reactivation of latent TB infection acquired more than two years earlier [12]. Research has shown that approximately 11% of the US-born persons assessed as having LTBI and around 12.4% of the non-US–born persons assessed as having LTBI had other associated medical risk factors [12]. The most common risk factor was diabetes without other concomitant conditions, such as ESRD or HIV, found in 8% of USA-born LTBIs and 10.7 % of non-USA-born LTBIs. This is followed by immunosuppressive therapy, which is responsible for 1.7% of LTBIs in US-born and 1% in non-USA-born individuals [12]. 

Tuberculosis Site: The TB site can be pulmonary or extrapulmonary. Normally, the upper lobes of the lungs are more frequently affected by Mycobacterium TB than the lower lobes. Extrapulmonary infections happen outside the lungs. This type of infection occurs commonly in people with weakened immunity and young children [13].

Pulmonary TB: TB infections that are inactive often involve the lungs. They constitute up to 90% of inactive infections. Symptoms include sticky sputum, prolonged coughing, and chest pain. About 30% of people remain asymptomatic; they cough up blood in small amounts, and in rare cases, the infection erodes into a Rasmussen’s aneurysm or pulmonary artery, leading to massive bleeding [14].

Extra-pulmonary TB: Extra-pulmonary TB occurs outside of the lungs and is a most likely sign of underlying immunological dysfunction. In most cases, it constitutes 10–20% of all active cases. This type of infection occurs commonly in people with weakened immunity and young children [13]. It happens mostly in people with HIV, where infection rates reach as high as 50% of cases. Notable sites for this disease include the central nervous system, TB pleurisy, the genitourinary system, the lymphatic system, joints, and bones.

Multiple Drug-Resistant TB [MDR-TB]: MDR-TB is an organism that is resistant to Isoniazid and Rifampin. Tuberculosis is treated with the use of multiple antibiotics for a longer period. However, antibiotic resistance is an emerging challenge [15]. There is a rise in extensively drug-resistant and MDR-TB. The resistance of Mycobacterium tuberculosis (MTB) to chemotherapeutic drugs was noticed immediately after the discovery of streptomycin in 1944 [16]. The spread of drug-resistant (DRTB), particularly multidrug-resistant MTB MDR-TB, has several consequences.

Statistical Analysis: Bivariate analyses were performed on all independent variables by using dependent variables. A chi-square analysis was conducted using crosstabs and Pearson’s chi-square for the independent variables of age, gender, race, country of birth, long-term care facility, and being homeless within the past year by the dependent variables of previous TB infection, multidrug-resistant tuberculosis, and site of tuberculosis. The bivariate logistic regression models were performed to examine the association between TB outcomes with sociodemographic variables. The odds ratio (OR) and 95% confidence interval (CI) were estimated.

## 3. Results

A total of 679 TB patients (443 men and 236 women) from 2011 to 2020 were selected from the MSDH TB Control database. Out of these patients, 37 were not included in the study analysis due to their eligibility criteria. The mean age of the 642 selected TB patients was 46 ± 10 years; 68% were aged >45 years old. Overall, 418 (65.1%) were male, and 224 (34.9%) were female TB patients. A total of 403 (59.53%) were TB patients of Black ethnicity, while 274 (40.47%) were White. More than 90% of the TB patients were born in the US, followed by Mexico 27 (4.2%) and India 22 (3.42%). Of the TB patients, 70 (10.9%) were homeless within the past year, while 572 (89.1%) were not homeless, and 22 (3.4%) were living in long-term care facilities. The highest number of TB patients were unemployed, 260 (40.5%), followed by others, 186 (29.0%), and retired persons, 131 (20.4%). Moreover, 15% were addicted to excessive alcohol use, and 1% were addicted to drug injections (Table 1).

### 3.1. Study Design and Analysis Subsection

The analysis was conducted at the patient level via the de-identified member identification number. Bivariate analyses were performed on all independent variables by dependent variables. A chi-square analysis was conducted using crosstabs and Pearson’s chi-square for the independent variables. The bivariate logistic regression models were performed to examine the association between TB outcomes with sociodemographic variables. The odds ratio (OR) and 95% confidence interval (CI) were estimated.

### 3.2. RQ1: Is There an Association between Demographic Subgroups (Race, Age, Country of Birth, and Gender) and Previous TB in Mississippi?

From Table 2, an analysis of the findings from summary statistics showed that the age class of 46–60 years had the highest rate of previous TB (31.9%) than other age subgroups; however, Pearson chi-square did not indicate any statistically significant difference (χ^2^ = 2.88, *p* = 0.57). Likewise, gender did not have any statistically significant difference (χ^2^ = 0.001, *p* = 0.97) by previous TB, among which 65.3% were males and 34.7% were females. However, the race factor showed statistical significance (χ^2^ = 4.55, *p* = 0.03) by previous TB, among which Black patients had a significantly higher (70.8%) previous TB history than White patients (29.2%). The results of the summary statistics also indicated that the difference in the country of birth of TB patients was not significant by previous TB (χ^2^ = 2.22, *p* = 0.13), and it was revealed that >87% of patients with previous TB were born in the US.

Based on the results from bivariate logistic regression in Table 2, older TB patients (TB patients with a higher age group compared with age group 18–30) were more likely to be associated with previous TB, but the relationship was not statistically significant (OR > 1, χ^2^ = 2.91, *p* = 0.59). Female patients were less likely to be associated with previous TB, which was also not statistically significant (OR = 0.99, χ^2^ = 0.001, *p* = 0.97). In addition, patients not born in the US were less likely to be associated with previous TB, which was not statistically significant (OR = 0.58, χ^2^ = 2.18, *p* = 0.13). However, race showed statistical significance with previous TB; the odds of previous TB among Black patients was 1.78 times higher compared to White patients (OR = 1.78, 95%CI =1.4–3.4, *p* = 0.03).

As a result, the logistic regression analysis for Hypothesis 1 rejected the null hypothesis for race only with statistically significant evidence and accepted the alternative hypothesis. There was a significant association between race and previous TB, i.e., the odds of previous TB among Black patients were 1.78 times higher compared to White patients. Further, the 95% CI for the OR was 1.04–3.04. The association between country of birth and previous TB was not statistically significant (OR = 0.58, 95%, CI =0.28–1.20, *p* = 0.13). 

### 3.3. RQ 2: Is Race Associated with TB Outcome Variables (MDR-TB and TB Site) Adjusting for Other Covariates?

A bivariate logistic regression model was performed to determine the comparative association, control the confounders, and provide composite effect estimates adjusted between categorical demographic variables with TB site and MDR-TB (Table 3). 

Based on the results from bivariate logistic regression in Table 3, Black patients were more likely to be associated with MDR-TB and less likely to be associated with pulmonary site TB, but both relationships were not statistically significant (OR = 1.02, 95% CI= 0.54–1.92, *p* = 0.96), (OR = 0.87, 95% CI = 0.55–1.36 *p* = 0.53).

As a result, the logistic regression analysis for Hypothesis 2 failed to reject the null hypothesis. Therefore, it was concluded that there was no association between TB outcome variables (MDR-TB, TB site) and race from 2011 to 2020 in Mississippi. From Table 4, summary statistics indicated that the race factor did not show any statistical significance (χ^2^ = 0.002, *p* = 0.96; χ^2^ = 0.39, *p* = 0.53) by both TB outcomes (MDR-TB and pulmonary site), among which Black patients were higher (59.5% and 58.7%) with both TB outcomes than White patients (40.5% and 41.3%).

### 3.4. RQ 3: Are There Differences between Homeless and Long-Term Care Facility Residents among TB Outcomes in Mississippi?

The frequencies and percentages of homelessness in the past year and people living in long-term care facilities among MDR-TB, previous TB, and TB-site were calculated using cross-tabulation and chi-square estimation to indicate the significant differences between long-term care facility and homeless residents’ diagnoses of TB. From Table 5, summary statistics indicated that 2.8% of patients of previous TB were in long-term care facilities, but the difference did not show any statistically significance (χ^2^ = 0.10, *p* = 0.74) by previous TB. However, homeless populations within the past year factor showed statistical significance (χ^2^ = 4.27, *p* = 0.03) by previous TB, amongst which 18.1% of patients with previous TB infections were homeless within the past year. For TB outcome of MDR-TB, the long-term care facility factor indicated statistical significance (χ^2^ = 5.04, *p* = 0.02), as 9.5% of patients of MDR-TB were residents of long-term care facilities. However, 5.8% of patients of MDR-TB were found to be homeless within the past year, which did not show any statistically significant difference (χ^2^ = 1.74, *p* = 0.18). For TB outcome with a pulmonary site, the long-term care facility factor did not show any statistical significance (χ^2^ = 2.81, *p* = 0.09), as 2.5% of patients with a pulmonary TB site were in a long-term care facility. However, 12.1% of patients with a pulmonary TB site were found to be homeless within the past year, which indicated a statistically significant difference (χ^2^ = 5.14, *p* = 0.02) by TB outcome.

Based on the outputs from logistic regression in Table 4, the statistically significant results indicated that those who were homeless within the past year were more likely to be associated with previous TB infection (OR = 1.98, 95%, CI = 1.03–3.84, *p* = 0.04), i.e., the odds for previous TB among those who were homeless within the past year was 1.98 times higher compared to those who were not homeless. 

The long-term care facility factor was significantly associated with MDR-TB (OR = 3.40, 95%, CI = 1.10–10.56, *p* = 0.03), i.e., the odds for MDR-TB among long-term care facility residents was 3.40 times higher compared to non-residents. Finally, there was significant association between being homeless within the past year and site of pulmonary TB (OR = 3.12, 95%, CI = 1.11–8.77, *p* = 0.03), i.e., the odds of pulmonary site TB among those who were homeless within the past year was 3.12 times higher compared to those who were not homeless. There was no statistically significant association between long-term care facility and previous TB (OR = 0.79, OR = 0.18–3.43, *p* = 0.74), as well as no statically significant association between long-term care facility and pulmonary site TB (OR = 0.45, 95%, CI = 0.17–1.17, *p* = 0.20). The association between homelessness and MDR-TB was not statistically significant (OR = 0.39, 95%, CI= 0.09–1.66, *p* = 0.20).

As a result, the logistic regression analysis for Hypothesis 3 rejected the null hypothesis with statistically significant evidence and accepted the alternative hypothesis. Therefore, we concluded that TB outcomes in Mississippi differ between those who are homeless and those in long-term care facilities.

## 4. Discussion

The purpose of this study was to assess the association between sociodemographic characteristics and TB outcome variables in Mississippi. The analysis of the findings showed that, among the sociodemographic factors, age, gender, and place of birth did not have a statistically significant association with previous TB. However, the race/ethnicity factor showed statistically significant association with previous TB, among which Black patients had a significantly higher previous TB history (70.8%) than White patients (29.2%). (OR = 1.78, *p* = 0.03). Nevertheless, Black patients were more likely to be associated with multiple drug-resistant TB and less likely to be associated with pulmonary site TB, but both relationships were not statistically significant. People who were homeless within the past year were more likely to have previous TB infection (OR = 1.98), and people who were homeless were more likely to be associated with pulmonary site TB (OR = 3.12). The long-term care facility factor was more likely to be associated with MDR-TB (OR = 3.40). These findings explain the disparity among the TB rates in Mississippi and specifically in counties, including Hinds County, as well as among the Black and White populations observed in this study. 

Black/African Americans have a significantly higher history of previous TB than the White population in Mississippi (Table 3). These results coincide with the CDC’s national TB report of 2020, which indicates that the rate of TB infection in Black people is eight times higher than for White Americans, with a 3.4 incidence rate. Several challenges associated with Black or African Americans may be contributing to these high rates of TB infection. These challenges include lengthy treatment, patients’ reluctance to take medication for several months, inadequate treatment, genetic complexities, latent TB infection, higher unemployment, poor housing and transportation conditions, and cultural barriers including health literacy, beliefs and other stigma associated to TB infection among certain populations [17,18,19,20].

In contrast, age and place of birth of TB cases in Mississippi did not show statistical significance (*p* > 0.05) concern with previous TB. The anticipated reason for this disparity is in the host and environmental factors, as well as comorbid conditions such as HIV or cancer [18]. The bivariate logistic regression model showed that the race/ethnicity factor did not show a statistically significant association with TB infection pulmonary site and MDR-TB (*p* > 0.05); this led to failure to reject the null hypothesis and the conclusion that there is no association between race and MDR-TB and TB pulmonary site.

The race/ethnicity demographic is presently considered a significant risk factor for TB around the world. Prior research has shown that the race/ethnicity risk factor is commonly associated with the incidence of TB in the United States [21,22]. African Americans are more related to the TB outcome pulmonary site, and this apparent difference in TB infection susceptibility could be a result of the specific response of African Americans’ genotype and comorbid conditions. These results are consistent with the outcomes of other studies, which have reported a higher risk of recent TB transmission in African Americans [19,23,24,25]. Although 37% of Mississippians are of African American descent, they count for 60% of all TB cases in the state. The recent investigation reported by [26] suggested that high neutrophil infiltration into TB granulomas is significantly associated with higher bacterial load, while the percentage of neutrophils among African Americans is averagely higher than other ethnic groups, which could be a possible reason for African Americans’ association with previous TB and pulmonary sites.

Even with an appreciation of the point that African Americans might be more susceptible to TB because of their response to TB infections and their genotype, the sociodemographic susceptibility of African Americans cannot be undermined. In 2017, the NCHS report stated that despite significant progress in reducing the gap in health outcomes, still, disparities in health outcomes exist for racial/ ethnic minorities [27]. Especially for the African American population, the extent of disparity for infectious diseases-related outcomes, such as HIV and TB, has increased substantially over the years [28]. 

Earlier research by [6] has also suggested that men have a higher likelihood of being diagnosed with TB compared to women. The predictors of greater susceptibility to TB infection and delay in diagnosis have been reported for both genders: for example, low socioeconomic status has been recorded as a barrier for some women to gaining access to the diagnosis and treatments for their TB disease worldwide [9, 27]. On the other hand, ref. [21,29] described HIV infection and indoor air pollution as the most common risk factors for transmitting TB among women. In contrast to women, men have faced different risk exposures for contracting and developing TB infection. As observed in this study, the social lifestyle of some men may be responsible for their exposure to TB [21]. 

In terms of sub-objectives, to our knowledge, this was the first study in Mississippi regarding MDR-TB with the aim of determining its association with sociodemographic factors. Only the long-term care facility factor was found to be significantly associated (OR = 3.40, 95% CI: 1.10–10.56, *p* = 0.03) with the occurrence of MBR-TB. For African Americans with TB disease, latent TB infection with no symptoms, poverty, and limited access to quality healthcare facilities may not be a priority to continue treatment, which causes the development of multidrug resistance among TB patients [5]. 

In the third research question of this study, we observed a significant difference (*p* < 0.05) among the TB patients who were living in long-term care facilities and homeless patients with previous TB. The patients who were homeless were significantly associated (OR = 1.98, 95% CI 1.03–3.84, *p* = 0.04) with previous TB, while the patients who resided in long-term care facilities were not significantly associated (*p* > 0.05) with previous TB. The Mississippi Morbidity Report [30] showed a link between the rise in TB in Hinds County with homeless TB-infected people. The high prevalence of previous TB among homeless TB-infected persons may be due to higher exposure to an open environment. The major outbreak contributors to TB in the homeless population were the lack of periodic screening in shelter clients, the deficiency of preventive treatment compliance, and the difficulty of follow-up, especially in subclinical TB infection [8].

Homeless case characteristics in prior research are comparable to our study population, including young adult males, a high prevalence of alcohol and drug abuse, and being born in the USA. The elderly population represents the largest reservoir of TB infection; an analysis of 1993–2008 cases reported in the United States showed that the rate of TB among elderly adults was as much as 30% higher than among younger adults [31]. Research has also shown that people over 65 years of age residing in long-term care facilities are at a 4–50-times higher risk of developing TB disease and experiencing reactivation of latent TB than elderly persons living in the community [31,32]. 

Diagnosing TB infections in the elderly population due to subtle clinical manifestations poses unique challenges, and it might also explain the observation in this study that patients who reside in long-term care facilities are not significantly associated (*p* > 0.05) with previous TB. In elderly individuals, typical presenting features of pulmonary TB disease, such as weight loss, cough, hemoptysis, and night sweats, may be either absent or attributable to alternative diagnoses. Moreover, false-negative tuberculin skin test (TST) results are more common in elderly populations due to impaired immunity leading to the possibility of unrecognized TB infection. In people living in long-term care facilities, there are other administrative barriers, such as low staff numbers, fewer resources for diagnosis and treatment, and physicians seeing more patients in less time. Additionally, there is a high turnover rate among long-term care facilities resulting in less trained and less experienced personnel taking care of these patients, leading to missed diagnoses and the mismanagement of TB cases [31]. 

## 5. Conclusions

The findings of this study suggest that sociodemographic factors play an important role in previous TB, MDR-TB, and TB infection sites; homelessness is also validated as a crucial covariate associated with cases of TB, as well as previous TB and the recurrence of cases in the study population. This research may lead to a constructive societal shift, by informing public health professionals in the development of an effective TB intervention that addresses sociodemographic and therapy-related risk factors for TB in Mississippi.

Future research targeting the control and eradication of TB in low-burden areas such as Mississippi should identify the differential effects of sociodemographic factors and eomoropid factors on TB infections, as well as how these factors differ for recently transmitted cases versus those resulting from the reactivation of latent TB infection (LTBI). These will assist in reducing risks and improving treatment. 

## Figures and Tables

**Table 1 diseases-11-00025-t001:** Sociodemographic characteristics of tuberculosis cases in Mississippi in 2011–2020.

Characteristic	Frequency (*n* = 642)	Percent 100%
Age, year		
18–30	74	11.5
31–45	133	20.7
46–60	198	30.8
61–75	141	21.9
>75	96	14.9
Gender		
Female	224	34.8
Male	418	65.1
Race		
Black	380	59.2
White	262	40.8
Country of birth		
Non-US born	122	19.0
US-born	520	81.0
Excess alcohol use		
No	547	85.2
Yes	95	14.8
Homeless within past year		
No	572	89.1
Yes	70	10.9
Long-term care facility		
No	620	96.6
Yes	22	3.4
Drug use (injections)		
No	636	99.1
Yes	6	0.9
Primary occupation		
Unemployment	260	40.5
Retired	131	20.4
Non-seeking	43	6.7
Others	204	31.8
Residence		
Hinds County	151	23.5
Harrison County	27	4.2
Jackson County	25	3.9
Desoto County	23	3.6
Other Counties	399	62.1

**Table 2 diseases-11-00025-t002:** Bivariate and multivariate analysis between previous TB and demographics.

	Bivariate Analysis		Multivariate Analysis	
	Yes	No	OR	95% CI
Age, N (%)				
18–30	5 (6.9%)	69 (12.1%)		
31–45	114 (19.4%)	119 (20.9%)	1.62	(0.56, 4.70)
46–60	23 (31.9%)	175 (30.7%)	1.81	(0.66, 4.96)
61–75	20 (27.8%)	121 (21.2%)	2.28	(0.82, 6.35)
>75	10 (13.9%)	86 (15.1%)	1.61	(0.52, 4.91)
Test statistics	2.88 *		2.81 **	
*p* value	0.57		0.58	
Gender				
Male	47 (65.3%)	371 (65.1%)		
Female	25 (34.7%)	199 (34.9%)	0.99	(0.59, 1.66)
Test statistics	0.001 *		0.001 **	
*p* value	0.97		0.97	
Race, N (%)				
White	21 (29.2%)	241 (42.3%)		
Black	51 (70.8%)	329 (57.7%)	1.78	(1.04, 3.04)
Test statistics	4.55 *		4.45 **	
*p* value	0.03		0.03	
Country of birth, N (%)				
US born	63 (87.5%)	457 (80.2%)		
Non-US born	9 (12.5%)	113 (19.8%)	0.58	(0.28, 1.20)
Test statistics	2.22 *		2.18 **	
*p* value	0.13		0.13	

Note: * *p* value indicated statistical significance at 95% confidence interval (*p* < 0.05) for Bivariate; ** *p* Value indicated statistical significance 95% confidence interval (*p* < 0.05) for Multivariate.

**Table 3 diseases-11-00025-t003:** Bivariate and multivariate analysis between race and outcome variables (MDR and TB-site).

	Bivariate Analysis		Multivariate Analysis	
	Yes	MDR No	OR	95% CI
Race, N (%)				
White	17 (40.5%)	245 (40.8%)		
Black	25 (59.5%)	355 (59.2%)	1.02	(0.54, 1.92)
Test statistics	0.002 *		0.002 **	
*p* value	0.96		0.96	
Race, N (%)		TB-Site Pulmonary		
White	226 (41.3%)	36 (37.9%)		
Black	321 (58.7%)	59 (62.1%)	0.87	(0.28, 1.20)
Test statistics	0.39 *		0.39 **	
*p* value	0.53		0.53	

Note: * *p* value indicated statistical significance at 95% confidence interval (*p* < 0.05) for Bivariate; ** *p* Value indicated statistical significance 95% confidence interval (*p* < 0.05) for Multivariate.

**Table 4 diseases-11-00025-t004:** Bivariate logistic regression of TB Outcome variables associated with living in long-term care facilities and homelessness.

Parameters	Model Estimate		Chi-Square	*p* Value *
	OR	95% CI		
TB outcome = previous TB infection				
Long-term care facility			0.10	0.74
No	REF			
Yes	0.79	(0.18, 3.43)		
Homeless within past year			4.13	0.04
No	REF			
Yes	1.98	(1.03, 3.84)		
TB outcome = MDR-TB (Multiple Drug-Resistant TB)				
Long-term care facility			4.49	0.03
No	REF			
Yes	3.40	(1.10, 10.56)		
Homeless within past year			1.62	0.20
No	REF			
Yes	0.39	(0.09, 1.66)		
TB outcome = TB Site (Pulmonary Site)				
Long-term care facility			2.67	0.10
No	REF			
Yes	0.45	(0.17, 1.17)		
Homeless within past year			4.65	0.03
No	REF			
Yes * Logistic regression; OR; CI	3.12	(1.11, 8.77)		

* *p* value indicated statistical significance at 95% confidence interval (*p* < 0.05).

**Table 5 diseases-11-00025-t005:** Association between living in long-term care facilities and homelessness, and TB outcome among the TB cases in Mississippi.

Parameters	TB Outcome		Chi-Square	*p* Value *
	Yes	No		
TB outcome = previous TB infection				
Long-term care facility, N (%)			0.10	0.74
No	70 (97.2%)	550 (96.5%)		
Yes	2 (2.8%)	20 (3.5%)		
Homeless within past year, N (%)			4.27	**0.03**
No	59 (81.9%)	513 (90.0%)		
Yes	13 (18.1%)	57 (10.0%)		
TB outcome = MDR-TB (Multiple Drug-Resistant TB)				
Long-term care facility, N (%)			5.04	**0.02**
No	38 (90.5%)	582 (97.0%)		
Yes	4 (9.5%)	18 (3.0%)		
Homeless within past year, N (%)			1.74	0.18
No	40 (95.2%)	532 (88.7%)		
Yes	2 (5.8%)	68 (11.3%)		
TB outcome = TB Site (Pulmonary Site)				
Long-term care facility, N (%)			2.81	0.09
No	531 (97.1%)	89 (93.7%)		
Yes	16 (2.9%)	6 (6.3%)		
Homeless within past year, N (%)			5.14	**0.02**
No	481 (87.9%)	91 (95.8%)		
Yes	66 (12.1%)	4 (4.2%)		

* Bolded *p* value indicates statistical significance at (*p* < 0.05).

## Data Availability

The data are available by request from the Mississippi State Department of Health. The data would be considered properly de-identified under HIPAA regulations (45 CFR 164.514(b)(2)).

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
