# Peer review of "Association of Sociodemographic Factors with Tuberculosis Outcomes in Mississippi"

_diseases, 2023, doi:10.3390/diseases11010025_

Round 1

Reviewer 1 Report

Association of Sociodemographic Factors with Tuberculosis 2 Outcomes in Mississippi

In this study, the authors intended to employed the data from TB patients from Mississippi Department of Health (2011-2020) and intended to examine the association between sociodemographic subgroups with TB outcome variables. This is an interestinng study, however the manuscript would benefit from the following changes/amendments:

1. Table 1. It would be worthwhile to present Sociodemographic characteristics vs gender, and calculating the significance of random distribution. 

2. Please give the inclusion and exclusion criteria. 

3. It is important to mention the co-morbidities in the TB subjects. 

4. Please mention how OR and logistic regression were calculated. 

5. It is not clear why the authors employ Pulmonary and Extra-Pulmonary categories in the analyses. 

6. Please elaborate the Abbreviation SES in its first use. 

Author Response

Table 1. It would be worthwhile to present Sociodemographic characteristics vs gender, and calculating the significance of random distribution. 

It was clear on table one.

Please give the inclusion and exclusion criteria. 

Inclusion Criteria: The participants fulfilled the following inclusion criteria, i.e., 18 years of age or older, TB confirmed cases.

Exclusion Criteria: Individuals having other Mycobacterial infections and aged less than 18 years will be excluded from the study.  

It is important to mention the co-morbidities in the TB subjects.

Diabetes, malnutrition, HIV, tobacco and alcohol use, and Chronic lung diseases. 

Please mention how OR and logistic regression were calculated. 

It is not clear why the authors employ Pulmonary and Extra-Pulmonary categories in the analyses. 

TB site either pulmonary or extra-pulmonary is one of the outcome variables. It is a good indicator of the patient’s immunity status, comorbidity, and mortality. 

Please elaborate on the Abbreviation SES in its first use. 

Done

Reviewer 2 Report

It is an important study on neglected, underestimated, and re-emerging infection. I have some inputs to improve the paper, mainly to enhance the method, and to rearrange the results.

In the Methods:

It is needed more information about how many tuberculosis case were at the MSHD registries, and how many of them was meeting the inclusion criteria, what were the exclusion reasons of the cases who didn’t meet the inclusion criteria. Also 1) the analysis, statistical package etc. 2) ethical procedure should be noted.

In the Results:

There are some repeats in the results, in the first and second paragraph such as: 68% were aged >45 years old, gender, ethnicity etc.

Why it was shown both bivariate and logistic regression for RQ1 and RQ2, but only LR for RQ3? I strongly suggest to the authors to merge Tb2 and Tb3, and also Tb 5 and Tb 5. It also prevents the confusion due to the tables’ titles.

In the Discussion:

No need to write all the values of chi-squares and p values in the discussion.

A limitation part should be added in either method or discussion section.

The results allow a discussion about the interaction of the factors such as people with color, homeless etc. The authors may dive deeper that.

Author Response

It is needed more information about how many tuberculosis cases were at the MSHD registries, and how many of them was meeting the inclusion criteria, what were the exclusion reasons of the cases who didn’t meet the inclusion criteria. Also 1) the analysis, statistical package etc. 2) ethical procedure should be noted.

Exclusion was due to an age of less than 18.

In the Results:

There are some repeats in the results, in the first and second paragraph such as: 68% were aged >45 years old, gender, ethnicity etc.

We corrected the repetitions.

Why it was shown both bivariate and logistic regression for RQ1 and RQ2, but only LR for RQ3? I strongly suggest to the authors to merge Tb2 and Tb3, and also Tb 5 and Tb 5. It also prevents the confusion due to the tables’ titles.

We merged Tables 2 and 3. And 4 and 5.

In the Discussion:

No need to write all the values of chi-squares and p values in the discussion.

A limitation part should be added in either method or discussion section.

The results allow a discussion about the interaction of the factors such as people with color, homeless etc. The authors may dive deeper into that.

We deleted most of the p and chi-squares in the discussion section.

Limitation Added to the Methods section.

Round 2

Reviewer 2 Report

Ethical approval is still not existing in anywhere in the manuscript. The ethical approval should be documented even this a registry-based study.

2. Statistical analyses should be summarized under the method section even though mentioned in the results section. 

3. In the logistic regressions, there are chi-square values but chi-square values are not calculated in logistic regressions. Please indicate which statistical package you applied.

4. This is descriptive study rather than "cross-sectional analysis of secondary data"

5. In the first round, I commented that “There are some repeats in the results, in the first and second paragraph such as: 68% were aged >45 years old, gender, ethnicity etc.”

The authors corrected only the age reputation that I gave as an example. There are still reputations on gender, ethnicity, homeless status etc which exit in both paragraphs. I recommend to the authors to read carefully both first and second paragraph and then edit.

6. Please see the attached file including my suggested format for the Table 2 and Table 3.

Author Response

Ethical approval is still not existing in anywhere in the manuscript. The ethical approval should be documented even this a registry-based study.

Ethical approval: The study was approved by the Institutional Review Board of JACKSON STATE UNIVERSITY (protocol code 0023-22 on June 17, 2022). 

  1. Statistical analyses should be summarized under the method section even though mentioned in the results section. 

Statistical Analysis: Bivariate analyses were performed on all independent variables by dependent variables. Chi-square analysis was conducted using crosstabs and Pearson’s chi-square for the independent variables of age, gender, race, country of birth, long-term care facility, and homeless within the past year by the dependent variables of previous TB infection, multi-drug-resistant tuberculosis, and site of tuberculosis. The bivariate logistic regression models were performed to examine the association between TB outcomes with socio-demographic variables. The odds ratio (OR) and 95% confidence interval (CI) were estimated

  1. In the logistic regressions, there are chi-square values but chi-square values are not calculated in logistic regressions. Please indicate which statistical package you applied. We use SPSS
  2. This is descriptive study rather than "cross-sectional analysis of secondary data"

Corrected

  1. In the first round, I commented that “There are some repeats in the results, in the first and second paragraph such as: 68% were aged >45 years old, gender, ethnicity etc.”The authors corrected only the age reputation that I gave as an example. There are still reputations on gender, ethnicity, homeless status etc which exit in both paragraphs. I recommend to the authors to read carefully both first and second paragraph and then edit.                                  It has been reviewed and corrected
  1. Please see the attached file including my suggested format for the Table 2 and Table 3.We changed tables 2 and 3 following the suggested format.
